# Plant-Assisted Synthesis of Ag-Based Nanoparticles on Cotton: Antimicrobial and Cytotoxicity Studies

**DOI:** 10.3390/molecules29071447

**Published:** 2024-03-23

**Authors:** Ana Krkobabić, Maja Radetić, Andrea Zille, Ana Isabel Ribeiro, Vanja Tadić, Tatjana Ilic-Tomic, Darka Marković

**Affiliations:** 1Faculty of Technology and Metallurgy, University of Belgrade, Karnegijeva 4, 11120 Belgrade, Serbia; anakrkobabic@gmail.com (A.K.); maja@tmf.bg.ac.rs (M.R.); 2Centro de Ciência e Tecnologia Têxtil (2C2T), Universidade do Minho, 4800-058 Guimarães, Portugal; azille@det.uminho.pt (A.Z.); afr@2c2t.uminho.pt (A.I.R.); 3Institute for Medical Plant Research “Dr Josif Pančić”, Tadeuša Košćuška 1, 11000 Belgrade, Serbia; vtadic@mocbilja.rs; 4Institute of Molecular Genetics and Genetic Engineering, University of Belgrade, Vojvode Stepe 444a, 11010 Belgrade, Serbia; tanja.ilictomic@gmail.com; 5Innovation Centre of the Faculty of Technology and Metallurgy, University of Belgrade, Karnegijeva 4, 11120 Belgrade, Serbia

**Keywords:** Ag nanoparticles, plant extract, cotton, antimicrobial, cytotoxicity

## Abstract

The syntheses of Ag-based nanoparticles (NPs) with the assistance of plant extracts have been shown to be environmentally benign and cost-effective alternatives to conventional chemical syntheses. This study discusses the application of *Paliurus spina-christi*, *Juglans regia*, *Humulus lupulus*, and *Sambucus nigra* leaf extracts for in situ synthesis of Ag-based NPs on cotton fabric modified with citric acid. The presence of NPs with an average size ranging from 57 to 99 nm on the fiber surface was confirmed by FESEM. XPS analysis indicated that metallic (Ag^0^) and/or ionic silver (Ag_2_O and AgO) appeared on the surface of the modified cotton. The chemical composition, size, shape, and amounts of synthesized NPs were strongly dependent on the applied plant extract. All fabricated nanocomposites exhibited excellent antifungal activity against yeast *Candida albicans*. Antibacterial activity was significantly stronger against Gram-positive bacteria *Staphylococcus aureus* than Gram-negative bacteria *Escherichia coli*. In addition, 99% of silver was retained on the samples after 24 h of contact with physiological saline solution, implying a high stability of nanoparticles. Cytotoxic activity towards HaCaT and MRC5 cells was only observed for the sample synthetized in the presence of *H. lupulus* extract. Excellent antimicrobial activity and non-cytotoxicity make the developed composites efficient candidates for medicinal applications.

## 1. Introduction

Nanotechnology has become an essential and inevitable research area in modern materials science [1]. Nanoscale dimensions of materials provide extraordinary physical, chemical, and biological properties due to the large surface area to volume ratio [2]. So far, much research has been conducted on the synthesis, characterization, and application of metal-based NPs. The unique catalytic, optical, magnetic, electrical, and antimicrobial properties of metal-based NPs make them intriguing for various biomedical, energy, or catalysis applications [1,3,4]. Numerous methods (laser ablation, radiolysis, γ-radiation, chemical reduction, sonochemical methods, microwave-assisted methods, etc.) have been established for the synthesis of metal-based NPs. Although very efficient, these methods involve chemical agents that are often toxic, polluting, and expensive [1,3,5,6,7].

Green syntheses of metal-based NPs have been proposed as environmentally friendly and cost-effective alternatives to conventional synthesis routes [8]. In green methods, biological microorganisms (bacteria, yeast, algae, enzymes, or fungi) or plant extracts act as reducing agents for metal ions. Compared to biological processes that require specific conditions (complex maintaining of cell cultures and necessity of sterile environments), plant extract-assisted syntheses of NPs are advantageous due to the abundance and low cost of plants [4,8,9,10,11,12,13]. In addition, plant extracts are rich in phytomolecules, such as water-soluble flavonoids, alkaloids, and phenolic compounds, widely known as polyphenols [14]. Polyphenols possess strong antioxidant activity, which enables them to act as reducing agents, hydrogen donors, and singlet oxygen quenchers. In addition, some of them have strong chelating ability [14,15]. Consequently, plant extracts act as reducing agents for metal ions and capping agents for metal NPs [4].

Among the metallic nanoparticles, silver (Ag) NPs have gained much attention due to their specific physical, biological, and chemical properties, including remarkable antimicrobial activity against various microorganisms such as bacteria, viruses, and fungi [16]. Green syntheses of Ag NPs using diverse plant extracts have been extensively explored over the last decade [8,14]. Different parts of plants, such as the roots (*Morinda citrifolia*, *Erythrina indica*), leaves (*Coffea arabica*, *Camellia sinensis*, *Eucaliptus globulus*, *Curcuma longa*), seeds (*Coriandrum sativum*, *Pistacia atlantika*), fruits (*Punica granatum*, *Sambucus nigra*), etc., can be utilized for preparing extracts for the synthesis of Ag NPs [4,8]. The present study discusses the possibility of using the leaf extract of *P. spina-christi* (Jerusalem torch), *J. regia* (walnut), *H. lupulus* (hop), and *S. nigra* (elder) for the synthesis of Ag-based NPs. The abundance in nature worldwide (Europe, North Africa, Middle East) of these plants was an additional reason to select them as green reducing agents [7,17]. Significant amounts of hop are being treated as a waste material in the brewing industry [8].

*J. regia* bark extract was used to synthesize Ag NPs for inhibition of *S. aureus* growth [1,18]. *J. regia* was also employed for the synthesis of Ag NPs to prevent the production of toxic aflatoxins by *Aspergillus ochraceus* [17]. The fruits of blackberries (*S. nigra*) could be utilized for plant-mediated synthesis of Ag NPs [6,7,19]. Das et al. synthesized antibacterial Ag NPs using hop extracts [8]. The focus of these reports was the synthesis of colloidal solutions of Ag NPs without providing deeper insight into particular applications. Our intention was to investigate the possibility of using plant-assisted in situ synthesis of Ag-based NPs on flexible substrate for achieving efficient antimicrobial activity. Namely, textile fibers naturally do not possess antimicrobial activity, and thus, antimicrobial finishing for applications in the medical and healthcare sectors is needed. So far, many different antimicrobial agents have been proposed (quaternary ammonia salts, n-halamine, triclosan, chitosan, dyes, etc.), but their efficiency and durability have often been lacking. On the other hand, the use of some of these was banned due to toxicity. Large surface-to-volume ratio and high reactivity have made metal and metal oxide NPs (Ag, Cu, CuO, Cu_2_O) promising agents for antimicrobial textile finishing. Small amounts of NPs can impart extraordinary antimicrobial activity to textile fibers using relatively simple processing routes. Keeping in mind that cotton is widely used in the fabrication of medical textiles and that the need for antimicrobial textiles is continually growing, cotton fabric is commonly used as a substrate for impregnation with biosynthesized Ag-based NPs [20,21,22,23,24,25]. In this work, a simple synthesis route without purification of plant extracts and without using any additional stabilizing agents is proposed. In situ biosynthesis of Ag-based NPs consists of three consecutive steps: (i) the preparation of the water extract of the selected plant, (ii) the sorption of Ag^+^-ions on cotton, and (iii) the reduction of Ag^+^-ions in plant extracts (synthesis of NPs). Cotton fabric was previously modified by citric acid to ensure enhanced sorption of Ag^+^-ions. The influence of the plant extract on the size and amount of synthesized NPs and consequently on the antimicrobial activity of the developed textile nanocomposites was tested against Gram-negative bacteria *E. coli*, Gram-positive bacteria *S. aureus*, and yeast *C. albicans*. The superiority of the fabricated nanocomposites is reflected in their low cytotoxicity to healthy human keratinocyte (HaCaT) and fibroblast (MRC5) cells.

## 2. Results and Discussion

### 2.1. Chemical Composition of Herbal Extracts

Generally, plant extracts are rich in various polyphenols, which are able to bioreduce Ag^+^-ions, resulting in the formation of Ag-based NPs [26]. However, the size, shape, agglomeration, yield, and time necessary for nanostructure growth are strongly dependent on the chemical composition of plant extracts, i.e., the reducing potential and stabilizing efficiency of the existing active molecules [27,28]. The reduction potential of plant polyphenols (E^0^[Ar–OH(I)/Ar–O^−^], where Ar represents the phenyl group) ranges from 0.3 to 0.8 V, making the polyphenol-assisted reduction of Ag^+^-ions (E^0^[Ag^+^/Ag] = 0.799 V) feasible [29]. Plant extract concentration, pH, and temperature also play important roles in the biosynthesis of Ag-based NPs [28]. Several recent studies have indicated that gallic acid acts as a reducing agent in the synthesis of Cu- and Ag-based NPs on cotton and viscose fabrics [24,30,31]. HPLC analysis of *P. spina-christi*, *J. regia*, *H. lupulus*, and *S. nigra* water extracts used in the current study revealed low concentrations of gallic acid (3.76, 1,20, 0.26, and 2.30 mg/g DW, respectively), indicating that some other molecules are also engaged in the reduction process (Table 1).

The major constituents of plant extracts are polyphenolic compounds. Total polyphenol increases in the following order: *H. lupulus* < *S. nigra* < *J. regia* < *P. spina-christi* extract. Large amounts of quercetin (3,3′,4′,5,7-pentahydroxyflavone)-polyphenolic flavonoid containing conjugated double bonds and five hydroxyl groups and its derivatives (rutin, hyperoside, isoquercetin) are found in *S. nigra*, *J. regia*, and particularly *P. spina-christi* extract. In contrast, the content in *H. lupulus* extract is very low. The obtained results imply that quercetin and its derivatives may be responsible for the reduction of Ag^+^-ions to Ag NPs [26,27,28,29]. It is assumed that hydroxyl groups in quercetin molecules are key actors in the reduction process. Namely, the tautomeric transformation of flavonoids from enol to keto form is followed by the release of reactive hydrogen, which can reduce Ag^+^-ions to Ag^0^. The catechol moiety in the quercetin structure may strongly chelate with Ag^+^-ions [32].

### 2.2. Chemical and Morphological Properties of the Samples

The exact mechanism of metal NP synthesis with plant extracts is not yet fully understood, but it is assumed that it is based on the following stages: reduction of metal ions and nucleation of reduced ions (activation phase) followed by clustering and further NP growth, where small neighboring NPs spontaneously merge into larger particles until they reach their final shape and size [26,33]. Such a mechanism explains the formation of colloidal metal NPs. In our case, the adsorption of Ag^+^-ions from the precursor salt solution precedes the described steps. Namely, in situ phytosynthesis was conducted on cotton fabric (CO) modified with citric acid (CA). Citric acid as a tricarboxylic acid can efficiently esterify hydroxyl groups of cellulose [34]. Free carboxyl groups have been proven to be sites for the sorption of copper ions [34]. Chemical changes induced by the modification of CO samples with citric acid along with the changes caused by subsequent impregnation with Ag-based NPs were evaluated by FTIR spectroscopy. The FTIR spectra of the CO, CO modified with citric acid (CO-CA), and CO modified with citric acid and Ag-based NPs in situ synthesized in the presence of leaf extracts of *P. spina-christi* (CO-CA-Ag-PSC), *J. regia* (CO-CA-Ag-JR), *H. lupulus* (CO-CA-Ag-HL), and *S. nigra* (CO-CA-Ag-SN) are shown in Figure 1. The bands characteristic for cellulose are clearly identified in all spectra: 3500–3200 cm^−1^ (the hydrogen-bonded O–H stretching in cellulose); 2898 cm^−1^ (C–H asymmetric stretching); 1630 cm^−1^ (adsorbed water molecule); 1427, 1364, 1315, and 1280 cm^−1^ (C–H in-plane bending vibrations, C–H bending (deformation stretch) vibrations, C–H wagging vibrations, and C–H deformation stretch vibrations, respectively); 1203 cm^−1^ (OH in-plane bending vibrations); 1156 and 1105 cm^−1^ (asymmetric bridge C–O–C); 1050 cm^−1^ (asymmetric in-plane ring stretching); 1031 cm^−1^ (C–O stretching) and 900 cm^−1^ (asymmetric out-of-phase ring stretching at C1–O–C_4_ β glucosidic bond) [35].

A new peak at 1720 cm^−1^ appearing in the spectrum of the CO-CA sample confirms the formation of ester bonds between hydroxyl groups of cellulose and carboxyl groups of citric acid [36,37,38,39]. It is reported that a cyclic anhydride is formed in the first step, which further establishes the ester bond with hydroxyl groups of cellulose [38,39]. After the synthesis of NPs, the intensity of peaks cantered at 1720 cm^−1^ were preserved, likely due to existence of carbonyl groups in keto forms of flavonoids [7]. Čuk et al. assumed that the biomolecules of the plant extracts are consumed in the NPs’ formation and adsorption on cellulose [40]. The intensity of this band decreased in the case of the CO-AC-Ag-HL sample, which be the consequence of a lower content of polyphenols in the *H. lupulus* extract (Table 1). A significant change in the shape of the peak cantered at 1640 cm^−1^ is observed in the spectra of all samples impregnated with NPs. This could be due to the formation of Ag-based NPs capped by molecules from the extracts [7,18,26]. The peak around 1640 cm^−1^ found in the plant extracts is ascribed to the carbonyl stretch of flavonoid derivatives [7,13,18,26].

XPS analysis of the CO sample and CO samples impregnated with Ag-based NPs (CO-CA-Ag-PSC, CO-CA-Ag-JR, CO-CA-Ag-HL, and CO-CA-Ag-SN) was performed to study the chemical modification of the fabric surface after the functionalization process and confirm the deposition and oxidation state of synthesized NPs. From the survey spectra, the most prominent peaks were identified (Figure 2). C1s and O1s bands were detected in all samples. The modified cotton samples showed additional bands corresponding to Ag3d (Figure 2). The elemental atomic percentages were also determined on the surface of the samples from the XPS survey spectra before and after the ion-sputtering cleaning cycle. The cleaning cycle was performed to remove the adsorbed aliphatic carbon on the surface and improve the intensity of the signals of the interest elements (Table 2 and Table 3, respectively). From the survey spectra, a higher concentration of Ag was detected on the surface of the CO-CA-Ag-HL (8.31%), CO-CA-Ag-PSC (6.44%), and CO-CA-Ag-JR (6.26%) samples after the cleaning process. A smaller amount of Ag was found in the CO-CA-Ag-SN sample (3.96%). Other elements were also detected in all treated samples: chlorine, magnesium, and calcium. Sulfur was only detected in CO-CA-Ag-SN, and phosphorus in CO-CA-Ag-PSC. Higher amounts of chlorine were found in the CO-CA-Ag-HL sample (3.82%). Overall, the results indicate that these minor elements were added during the textile functionalization and also come from plant extracts.

The C1s high-resolution spectra showed three typical peaks of cellulose materials in all samples at 284.8 eV, 286.4 eV, and 287.8 eV, attributed to C-C/C-H, C-O, and O-C-O groups (Table 4 and Figure 3). An additional peak at 289.9 eV emerged in CO-CA-Ag-JR and CO-CA-Ag-PSC, corresponding to C=O moieties [41,42], which is in line with the FTIR spectra. Namely, they are attributed to carbonyl groups in keto forms of flavonoids.

The O1s core level of the XPS spectrum presented a well-defined peak centered at 532.9 eV in all samples, which corresponds to the characteristic binding energy of the single-bonded oxygen from the alcohol groups and O-C-O groups in cellulose (Table 4 and Figure 4) [43]. An additional peak at 531.6 eV appears in the CO-CA-Ag-SN and CO-CA-Ag-HL samples, which is ascribed to carbonyl groups formed during the functionalization. Two new peaks emerged at 534.0 and 535.8 eV in the CO-CA-Ag-JR and CO-CA-Ag-PSC samples. The first one is attributed to the free carboxylic groups of citric acid and in the plant extracts. The second one, at 535.8 eV, corresponds to the water molecules adsorbed on the CO fabric [44,45].

The high-resolution spectra corresponding to Ag3d core energetic levels indicated the presence of different Ag species in the treated CO samples (Figure 5 and Table 5). A common doublet emerges at 367.9-368.4 eV (Ag 3d5/2) and 373.9-374.4 eV (Ag 3d3/2) in all samples, which is attributed to metallic silver or Ag(I) [46]. Due to the overlap between the Ag^0^ and Ag^+^ regions in the XPS, it was not possible to distinguish between them. The auger peaks, which could help in the identification, present a small intensity due to the low amount of Ag in the samples and thus do not allow the identification of the species [47]. A second doublet appears at 369.7–369.8 eV and 375.7 eV in CO-CA-Ag-JR and CO-CA-Ag-PSC samples. The last doublet is found in the CO-CA-Ag-JR sample at 371.4 eV and 377.4 eV. Those doublets are attributed to ionic Ag with higher oxidation states. The second doublet corresponds to AgO, and the third to the mixture Ag_2_O/AgO [47,48]. The intensity of the peaks confirmed the information of the survey spectra, showing a higher Ag concentration in CO-CA-Ag-PSC and CO-CA-Ag-HL samples.

The formation of Ag-based NPs on the CO-CA sample was confirmed by FE-SEM analysis. FE-SEM microphotographs of synthesized NPs on CO-CA fibers using *P. spina-christi*, *J. regia*, *H. lupulus*, and *S. nigra* extracts and corresponding size distribution of the synthesized NPs are shown in Figure 6. Obviously, the application of all tested plant extracts resulted in the formation of NPs on the fiber surface. Spherical, mostly single NPs with an average size of 63±18 nm are visible across the CO-CA-Ag-PSC fibers (Figure 6a). The size distribution (Figure 6b) indicates that the size of most NPs ranges between 50 and 60 nm. Figure 6c reveals evenly distributed single and agglomerated Ag-based NPs (66±17 nm) over the surface of the CO-CA-Ag-JR fiber. The presence of folds nearly parallel to the fiber axis, characteristic for CO fibers, is also noticeable in Figure 6c. Despite very low total polyphenol and flavonoid content in the *H. lupulus* extract (Table 1), the surface of CO-CA-Ag-HL is fully covered with NPs, with an average size of 68 ± 15 nm (Figure 6e). The morphology of the CO-CA-Ag-HL fiber surface differs from the other samples, as the majority of NPs are organized in small, evenly distributed agglomerates. This could be the reason for the different color of the fabric. The NPs in this case are not spherical. Figure 6g shows that application of *S. nigra* extracts resulted in the synthesis of bigger NPs (101 ± 24 nm). The agglomeration of NPs is also visible on the CO-CA-Ag-SN fibers. The smallest NPs were synthesized using *P. spina-christi* extract, which contains the highest concentration of polyphenols (493 ± 3.05 mgGAE/g ext.) and low concentration of flavonoids (2.13 ± 0.11%). At the same time, AAS measurements demonstrated that the CO-CA-Ag-PSC sample contains the largest amounts of silver (27.4 ± 1.0 µmol/g). The largest NPs were found on the CO-CA-Ag-SN sample. Table 1 reveals that the extract of *S. nigra* contains the highest concentration of total flavonoids (4.55 ± 0.1%) and quercetin (49.07 ± 1.02 mg/g DW). The total amount of Ag in the CO-CA-Ag-SN sample is 21.2±0.2 µmol/g. The syntheses of Ag-based NPs assisted by *J. regia* and *H. lupulus* extracts brought about the formation of NPs of almost identical average size, and these samples contain 17.2 ± 2.3 and 19.0 ± 0.1 µmol/g of Ag, respectively. The polyphenol and flavonoid contents were the lowest in the *H. lupulus* extract. It is well known that hydroxyl and carboxyl groups in polyphenolic compounds successfully bind metals [49] as well as that quercetin is involved in the stages of initiation of NP formation (nucleation) and its further aggregation [33]. Namely, quercetin is a flavonoid with very strong chelating activity through its carbonyl and hydroxyls groups at the C3 and C5 positions and the catechol group at the C3′ and C4′ positions [33]. Therefore, it can be adsorbed on the surface of nascent NPs. However, the results of AAS, total polyphenolic, and flavonoid content measurements imply that no clear relation between extract chemical composition and the formation of Ag-based NPs can be established. The same conclusion was reported by Čuk et al. [40].

### 2.3. Antimicrobial Activity of Textile Nanocomposites

An urgent problem in medicine today is hospital-acquired infections, particularly those caused by antimicrobial resistant pathogens [16]. Since textiles, as commonly used materials in hospitals, are involved in the spreading of pathogens, medical textiles should ensure an adequate level of antimicrobial protection. Cotton in contact with the human body becomes a suitable substrate for microbial growth and represents an additional source of pathogens [50]. This could be overcome by impregnation of CO materials with Ag NPs. Our preliminary study revealed that CO-CA and samples loaded only with herbal extracts did not exhibit any antimicrobial activity towards the investigated pathogens. Table 6 summarizes the results of the antimicrobial reduction of *E. coli*, *S. aureus*, and *C. albicans* in contact with synthesized nanocomposites. The results point out that impregnation with Ag-based NPs imparted high antimicrobial activity to CO fabrics independently of the applied plant extracts. All samples show particularly high antibacterial activity against *S. aureus* (R = 99.9%). Filipič et al. also reported that maximum *S. aureus* reduction (100%) could be achieved on cotton fabric with a sol-gel matrix and in situ synthesized Ag particles using sumac leaf extract [25]. Equivalent antimicrobial activity was obtained with cotton fabrics impregnated with Ag NPs in situ synthesized using green tea leaves, avocado seed, goldenrod flowers, and pomegranate peel extracts [40]. CO-CA-Ag-PSC and CO-CA-Ag-HL samples showed very similar antimicrobial activity. CO-CA-Ag-JR and CO-CA-Ag-SN samples provided maximum reduction of *S. aureus* (99.9%), satisfactory reduction of *E. coli* (99.2%), and excellent antifungal activity.

Compared to our previous results, where AgCl NPs were in situ synthesized on chitosan-modified viscose fabric using pomegranate peel extract (*Punica granatum*), the samples developed in this study showed higher antibacterial activity [31]. Higher antimicrobial activity is suggested to be due to the higher content of Ag: the total content of Ag after synthesis with pomegranate peel extract ranged from 8.8 to 11.9 μmol/g, while the samples fabricated in the current study contain between 17.2 and 27.4 μmol/g of Ag.

### 2.4. Release of Ag^+^-Ions from Textile Nanocomposites

Synthesized nanocomposites as potential medical textiles will be in direct contact with humans. Therefore, an understanding of the NPs’ properties and their effect on the body and body liquids is crucial. To simulate body fluids, physiological saline solution was selected as a release medium. Figure 7 presents the release kinetics of Ag^+^-ions from the nanocomposites in physiological saline solution within 24 h. The same amount of Ag (~0.3 μmol/g) was released in physiological solution from the CO-CA-Ag-PSC sample during the whole investigation period (Figure 7a). Taking into account the total Ag content in the CO-CA-Ag-PSC sample and the maximum amount of Ag^+^-ions released from the sample after 24 h, it was calculated that approximately 99% of Ag was retained in the sample. Almost double the amount (~0.65 μmol/g) of released ions was detected in the CO-CA-Ag-JR sample (Figure 7b), and consequently, around 96% of Ag remained in the sample after 24 h in the physiological solution. Figure 7c shows a completely different release profile as compared to the others. Namely, the CO-CA-Ag-HL sample showed a time-dependent release of Ag^+^-ions. Approximately 0.35 μmol of Ag^+^-ions leached out from 1 g of the CO-CA-Ag-HL sample after 1 h, and 0.75 μmol/g after 24 h. The CO-CA-Ag-SN sample released a similar amount of Ag^+^-ions within 24 h. Around 0.35 μmol/g of Ag^+^-ions leached out from the CO-CA-Ag-SN sample, leaving almost 99% of Ag in the sample. The overall release of Ag+-ions in the present study was larger compared to the release from viscose fabric synthesized with pomegranate peel extract [29]; in addition, much larger amounts of Ag were retained in these samples.

### 2.5. Cytotoxicity of Textile Nanocomposites

Nanotoxicology research is continually growing, as nanostructures are widely used in different areas. Numerous studies emphasize the possible cytotoxicity of metal-based NPs. Keeping in mind that developed textile nanocomposites would be in direct contact with humans, the nature of their impact on skin must be explored in more detail. Skin consists of three types of cells: keratinocytes, melanocytes, and fibroblasts. The cell viability test was performed on human keratinocytes (HaCaT) and fibroblast cells (MRC5). According to ISO 10993-5 [51], when the percentage of cell viability exceeds 80%, the substrate is considered as non-cytotoxic; in the range between 60 and 80%, the cytotoxicity is weak; between 40 and 60%, it is moderate; and below 40%, it is considered strong [51]. The percentage of HaCaT and MRC5 cell survival after 24 h of contact with nanocomposite extracts in comparison with native cells (control) is presented in Figure 8. Evidently, only the highest concentration of CO-CA-Ag-HL extract causes a cytotoxic effect on HaCaT cells. Independently of the concentration of material extracts, the other samples did not induce any reduction of HaCaT (Figure 8a) and MRC5 (Figure 8b) cell growth. A significant reduction in MRC5 cell growth was observed in 50% and 100% of the material extract of the CO-CA-Ag-HL sample. According to ISO 10993-5, these samples show strong cytotoxicity. The other nanocomposites did not induce any reduction of MRC5 cell growth. The non-cytotoxicity of CO-CA-Ag-PSC, CO-CA-Ag-JR, and CO-CA-Ag-SN extracts towards HaCaT and MRC5 cells may be attributed to the high stability of these samples in the physiological system (Figure 7). Accordingly, these samples could be considered safe for further use.

Ribeiro et al. reported that PES samples with Ag NPs did not provide any cytotoxic effect towards HaCaT cells, independently of concentration [52]. Xu et al. also suggested that cotton fabric modified with L-cysteine and Ag NPs did not leach species toxic to HaCaT cells within 24 h [53]. Ballottin et al. proposed that the IC50 for Ag NPs in the case of a standard fibroblast cell line (3T3) and lymphocytes was 39.51 µg/mL for both cells [54]. It was also reported that HaCaT cell viability was around 80% after being exposed to 10 µg/mL of Ag NPs coated with citrate [55].

## 3. Materials and Methods

### 3.1. Materials

Desized and bleached cotton (CO) fabric (117.5 g/m^2^, 27 picks/cm, 52 ends/cm, thickness of 0.26 mm, Tekstina d.d., Ajdovščina, Slovenia) was used as a substrate. Washing agent Felosan RG-N was supplied from CHT Switzerland AG (Montlingen, Switzerland). All other reagents of p.a. grade were used in the experiments without any further purification. The catalyst sodium hypophosphite (SHP) was supplied by ACROS Organics (Geel, Belgium). Citric acid, nitric acid, and sodium chloride were purchased from Zorka Pharma, Šabac, Serbia. Silver nitrate was provided by Centrohem, Stara Pazova, Serbia. Sodium hydroxide was purchased from Lachema, Brno, Czech Republic. Microbial inoculum was prepared in triptone soya broth, agar, and yeast extract provided by Torlak, Beograd, Serbia. The standards for quantitative and qualitative analysis were purchased form Extrasynthese, Genay, France (protocatechuic acid (purity (HPLC) ≥ 90%), vanillic acd (purity (HPLC) ≥ 95%), neochlorogenic acid (purity (HPLC) ≥ 99%), gallic acid (purity (HPLC) ≥ 99%), ellagic acid (purity (HPLC) ≥ 95%), chlorogenic acid (purity (HPLC) ≥ 99%), 3,5-di-*O*-caffeoylquinic acid (purity (HPLC) ≥ 97%), rutin (purity (HPLC) ≥ 99%), hyperoside (purity (HPLC) ≥ 98%), isoquercetin (purity (HPLC) ≥ 99%), miquelianin (purity (HPLC) ≥ 98%), quercitrin (purity (HPLC) ≥ 98.5%), kaempferol-3-*O*-glucoside (purity (HPLC) ≥ 99%), quercetin (purity (HPLC) ≥ 99%), kaempferol (purity (HPLC) ≥ 99%) and isorhamnetin (purity (HPLC) ≥ 99%)). Human fetal lung fibroblasts (MRC5) (CCL-171) and human epidermal keratinocyte (HaCaT) (PCS-200-011) cells were previously obtained from ATCC (Manassas, VA, USA). The plants *P. spina-christi Mill*. (*Rhamnaceae*), *J. regia L.* (*Juglandaceae*), *H. lupulus L.* (*Cannabaceae*), and *S. nigra L. (Adoxaceae*) were kindly supplied by the Institute for Medical Plant Research “Dr Josif Pančić”, Belgrade, Serbia. The voucher specimens (PSC_202210MR, SN_202210MR, JR_2022MR, HL_202210MR) were deposited at the Herbarium of the Faculty of Pharmacy, University of Belgrade (Belgrade, Serbia), where identity confirmation was performed.

### 3.2. Preparation of CO Fabrics

CO fabric was cleaned in a bath containing 0.1% nonionic washing agent Felosan RG-N at a liquor-to-fabric ratio of 50:1. After 15 min of washing at 50 °C, the fabric was rinsed first with warm water (50 °C) and then thoroughly with cold water. The samples were dried at room temperature.

### 3.3. Modification of CO Samples with Polycarboxylic Acid

The modification of CO samples with citric acid was performed according to the following method: 0.50 g of the sample was immersed in 20 mL of citric acid solution (10 *w*/*v*%); 2.06 g of catalyst sodium hypophosphite (SHP) was added. After 1 h, the sample was dried at 80 °C for 3 min and cured at 170 °C for 3 min. The samples were then rinsed in distilled water and dried at room temperature. CO samples modified with citric acid are denoted as CO-CA.

### 3.4. Preparation of Plant Extracts

The plants *P. spina-christi* (PSC, Jerusalem thorn), *J. regia* (JR, walnut), *H. lupulus* (HL, hop), and *S. nigra* (SN, elder) were selected for the synthesis of Ag-based NPs. The plant extracts were prepared by adding 5.00 g of leaves (previously milled) to 100 mL of distilled water. The vessel with an extract was sealed and heated at 80 °C for 2 h. Prepared extracts were filtered and immediately used for the synthesis of Ag-based NPs.

### 3.5. In Situ Synthesis of Ag-Based NPs on the Modified Samples

A total of 0.50 g of the CO-CA sample was immersed in 25 mL of 20 mM solution of AgNO_3_. After a 2 h sorption, the sample was rinsed two times with deionized water and put in the freshly prepared solution of plant extract. The synthesis step was carried out at 60 °C for 1 h. Afterwards, the samples were thoroughly rinsed with distilled water and left to dry at room temperature. The samples synthetized using PSC, JR, HL, and SN are denoted as CO-CA-Ag-PSC, CO-CA-Ag-JR, CO-CA-Ag-HL, and CO-CA-Ag-SN, respectively. The synthesis route is schematically presented in Figure 9. The color coordinates of impregnated samples are given in Appendix A.

### 3.6. HPLC Analysis of Plant Extracts

Determination of the identified compounds in the plant infusions prepared in the manner previously described was carried out by the 1200 HPLC system (Agilent Technologies, Santa Clara, CA, USA) equipped with the LiChrospher^®^ 100, an RP-18e (5 µm, 250 × 4) column using two mobile phases (phase A being 0.1 M solution of phosphoric acid, and phase B being pure acetonitrile), and applying the established HPLC method with slight modifications [56]. The flow rate was 0.800 mL/min, with photodiode-array (PDA) detection (UV at 260 nm), always within 50 min. Best peak separation was achieved using the following combination: 2% B (0 min); 2–10% B (0–5 min); 10% B (5–15 min); 10–15% B (15–20 min); 15–60% B (20–40 min); and 60–80% B (40–60 min). Prior to injection, the investigated extract (50.00 mg/mL) was filtered through a PTFE membrane filter. The injection volume of both standard solutions and tested extracts was 5 µL. For standards protocatechuic, vanillic, neochlorogenic, gallic, 3,5-di-*O*-caffeoylquinic, ellagic, and chlorogenic acids, rutin, hyperoside, isoquercetin, miquelianin, quercitrin, kaempferol-3-*O*-glucoside quercetin, kaempferol, and isorhamnetin, the concentrations were 0.53, 0.28, 0.30, 0.12, 0.39, 0.45, 0.48, 0.40, 0.39, 0.32, 0.52, 0.39, 0.49, 0.41, and 0.15 mg/mL, respectively. Identification was based on overlay curves and retention times. After successful spectra matching, the results were confirmed by spiking with the respective standard to achieve a complete identification using the so-called peak purity test, and the peaks that did not fulfil these requirements were not quantified. Quantification was performed by external calibration with a standard. Triplicate measurements were taken, and data were presented as mean ± standard deviation (SD).

### 3.7. Total Phenolic Content

The total phenolic content was determined by the Folin–Ciocalteu method [21]. One hundred microliters of methanol solution of the investigated sample (the starting concentration being 30.00 mg/mL) was mixed with 0.75 mL of Folin–Ciocalteu reagent (previously diluted 10-fold with distilled water) and allowed to stand at 22 °C for 5 min. A total of 0.75 mL of sodium bicarbonate (60 g/L) solution was added to mixture. After 90 min at 22 °C, absorbance was measured at 725 nm. Gallic acid (0–100 mg/L) was used for calibration of a standard curve. The calibration curve showed the linear regression at R^2^ > 0.99, and the results are expressed as milligrams of gallic acid equivalents per g dry weight (DW). Triplicate measurements were taken, and data were presented as mean ± standard deviation (SD).

### 3.8. Total Flavonoid Aglycones Content

The total flavonoid aglycones content after acid hydrolysis was determined by HPLC using a method described by [57]. Hydrolysis was performed using the following procedure: 4.00 g of investigated extract was treated with 0.824 g conc. HCl and 3 mL of MeOH at 85 °C and refluxed for 2 h. After adding 3 mL of MeOH and transferring the reaction mixture into a 10 mL volumetric flask, the reaction mixture was subjected to an ultrasonic bath for 10 min at room temperature before being brought to the volume using MeOH. The hydrolyzed mixture was filtered through a 0.2 µm PTFE filter, and the volume of 4 µL was subjected to HPLC analysis for determination of flavonoid aglycones content after acid hydrolysis (quercetin, kaempferol, and isorhamnetin). The determination was performed in triplicate. The HPLC method details are described in the previous session (see HPLC analysis of plant extracts).

The total flavonoid content was calculated using the method described in the European Pharmacopoeia 10.0 [58]. Briefly, the sample was extracted with acetone/HCl under a reflux condenser; the AlCl_3_ complex of flavonoid fraction extracted by ethyl acetate was measured by a HP 8453 UV-VIS spectrophotometer (Agilent Technologies, Santa Clara, CA, USA) at *λ*_max_ 425 nm. The content of flavonoids (mean of three determinations), expressed as hyperoside percentage, was calculated using the following expression:A × 1.25/m(1)
where A is absorbance at 425 nm and m is mass of the extracts to be examined, in grams.

### 3.9. FTIR Analysis

The Fourier transform infrared (FTIR) spectra of the CO sample, CO-CA, CO-CA-Ag-PSC, CO-CA-Ag-JR, CO-CA-Ag-HL, and CO-CA-Ag-SN samples were recorded in the ATR mode using an iS5 FTIR Spectrometer (Thermo Scientific, Waltham, MA, USA) at 2 cm^−1^ resolution, in the wavenumber range of 500–4000 cm^−1^.

### 3.10. FE-SEM Analysis

The morphology of fibers impregnated with Ag-based NPs was evaluated by field emission scanning electron microscopy (FE-SEM, Tescan Mira3 FEG, Brno, Czech Republic). The samples were coated with a thin layer of gold prior to analysis. To define the size and size distribution of NPs, the obtained FESEM images were analyzed using the open-access imaging software tool ImageJversion 1.8.0.

### 3.11. AAS Analysis

The total amount of Ag in all samples was measured using a Spectra AA 55 B (Varian, Palo Alto, CA, USA) atomic absorption spectrometer (AAS). CO samples impregnated with Ag-based NPs were dissolved in the 1:1 HNO_3_ solution. Additionally, AAS was employed for the assessment of the Ag^+^-ions’ release from samples into the physiological saline solution.

### 3.12. XPS Analysis

X-ray photoelectron spectroscopy (XPS) analysis was performed at the “servicio nanotecnología y análisis de superficies” at C.A.C.T.I., University of Vigo, Spain. The samples were analyzed using a Thermo Scientific NEXSA XPS instrument equipped with an aluminum-Kα monochromatized radiation (1486.6 eV) X-ray source. The residual vacuum in the X-ray analysis chamber was maintained at around 4.9 × 10^−8^ torr. An electron flood gun was used to minimize surface charging due to the non-conductor nature of the samples. Neutralization of the surface charge was performed by using both a low-energy flood gun (electrons in the range of 0 to 14 eV) and a low-energy Argon ion gun. The XPS measurements were carried out using monochromatic Al-Kα radiation (1486.6 eV). Photoelectrons were collected from a take-off angle of 90° relative to the sample surface. The measurement was performed in Constant Analyzer Energy mode (CAE) with 100 eV pass energy for the survey spectra and 20 eV pass energy for the high-resolution spectra. Charge referencing was performed by setting the lower binding energy C1s photo peak at the 284.80 eV C1s hydrocarbon peak. Surface elemental composition was determined using the standard Scofield photoemission cross sections. The residual vacuum in the analysis chamber was maintained at 3 × 10^−9^ mbar. The analysis was performed before and after an ion-sputtering cleaning cycle. The cleaning cycle was performed with Argon ion energy of 1000 V for 60 s. The intensity of the bands was estimated by calculating the area under peaks after the subtraction of the (Shirley) S-Shaped background. The experimental curve was fitted using a mix of Lorentzian–Gaussian lines in variable proportions. High-resolution narrow scans were used to build the chemical state assessment and to quantify the presence of the reference elements in the sample. This process has an associated error of ±0.1 eV. CASAXPS software (version 2.3.15) was used to analyze the spectra for elemental composition. Deconvolution into sub-peaks was performed by least-squares peak analysis software, XPSPEAK version 4.1, using the Gaussian/Lorentzian sum function and Shirley-type background subtraction. In the peak fitting procedure, no tailing function was considered. The components of the various spectra were mainly modelled as symmetrical Gaussian peaks. The chemical composition of the samples and environment were examined ex situ by XPS surface measurements, acquiring C1s, O1s, N1s, Ag3d, and survey spectra.

### 3.13. Antimicrobial Test

The antimicrobial activity of the samples with Ag-based NPs was tested against Gram-negative bacteria *E. coli* ATCC 25922, Gram-positive bacteria *S. aureus* ATCC 25923, and yeast *C. albicans* using a standard test method for determination of the antimicrobial activity of immobilized antimicrobial agents under dynamic contact conditions ASTM E 2149-01 (2001). Microorganism inoculums were cultivated in the triptone soya broth (3 mL) supplemented with 0.6% *w*/*v* yeast extracts at 37 °C and left for 18 h (late exponential stage of growth). Freshly grown microbial cultures were diluted in sterile physiological saline solution to obtain inoculum with an initial number of cells of ca. 10^6^ CFU/mL for *E. coli* and ca. 10^5^ CFU/mL for *S. aureus* and *C. albicans*. A total of 50 mL of sterile physiological saline solution (pH 7.2) was inoculated with 0.5 mL of a microorganism inoculum. One gram of the control CO sample or modified samples (previously sterilized by UV light for 30 min) was placed in the flask at 37 °C and shaken for 2 h. Then, 1 mL aliquots from the flask were diluted with physiological saline solution, and 1.0 mL of the solution was placed onto a tryptone soya agar supplemented with 0.6 *w*/*v*% yeast extracts. After 24 h of incubation at 37 °C, the zero time (initial number of bacteria colonies) and 2 h counts of viable bacteria were taken.

The percentage of bacterial reduction (R, %) was calculated in accordance with the following equation:(2)$$R=\frac{C_{0}-C}{C_{0}}\times 100$$
where C_0_ (CFU/mL − colony forming units) is the number of bacteria colonies on the control fabric, and C (CFU/mL) is the number of bacteria colonies on the fabric impregnated with Ag-based NPs. Antimicrobial tests were run in triplicate.

### 3.14. The Release of Ag^+^-Ions from the Samples into the Physiological Solution

Ag^+^-ion release was tested by immersing 0.25 g of the modified sample in 25 mL of physiological saline solution (9 g/L NaCl) at 37 °C in static conditions. The concentration of released Ag^+^-ions was measured after 1, 3, 6, and 24 h by Spectra AA 55 B (Varian) AAS. The experiments were performed in triplicate.

### 3.15. Cytotoxicity Test

The cytotoxicity of the samples with synthesized Ag-based NPs was tested on human keratinocyte cells (HaCaT cell line obtained from ATCC) and fibroblast cells (MRC5 cell line obtained from ATCC) (in accordance with the method previously described in [59]). The threads pulled out from the textile nanocomposite were soaked in RPMI-1640 medium (10 mg/mL) and incubated at 37 °C. After 72 h of shaking at 180 rpm, the suspensions were centrifuged for 10 min at 5000 rpm (Eppendorf Centrifuge 5804R, Hamburg, Germany), and the supernatants sterilized with a 0.22 μm filter (Millipore, Billerica, MA, USA) were used in different concentrations.

The cells were plated in a 96-well flat-bottom plate at a concentration of 1 × 10^4^ cells per well, grown in humidified atmosphere of 95% air and 5% CO_2_ at 37 °C, and maintained as monolayer cultures in RPMI-1640 medium supplemented with 100 µg/mL streptomycin, 100 U/mL penicillin, and 10% (*v*/*v*) fetal bovine serum (FBS). After 24 h of cell incubation, the media containing increasing concentrations of material extracts at 25%, 50%, and 100% (*v*/*v*) were added to the cells. Control cultures contained 200 µL of growth medium. After 24 h of incubation, cell cytotoxicity was determined using 3-(4,5-dimethylthiazol-2-yl)-2,5-diphenyltetrazoliumbromide (MTT) reduction assay. The extent of MTT reduction to formazan within cells was measured by absorbance at 540 nm on a Tekan Infinite 200 Pro multiplate reader (Tecan Group Ltd., Männedorf, Switzerland). The MTT assay was conducted twice with four replicates, and the results were presented as a percentage of the control (untreated cells), which was arbitrarily set to 100%. The results are presented as means ± SD.

## 4. Conclusions

The present study revealed that, independently of the total polyphenols and flavonoid content, *P. spina-christi*, *J. regia*, *H. lupulus*, and *S. nigra* leaf extracts could be successively used for the synthesis of Ag-based NPs on cotton textile material. FTIR analysis showed that modification with citric acid resulted in the introduction of carboxyl groups that act as active sites for the sorption of Ag^+^-ions. The total amount of Ag after the synthesis of NPs as well as the size and shape of NPs were dependent on the applied herbal extracts, but correlation with their chemical composition could not be established. The highest content of Ag (27 μmol/g) and the smallest particle size (57 nm) was observed on the sample synthesized in the presence of the *P. spina-christi* extract. However, the concentration of Ag in all samples was sufficient to provide excellent antimicrobial activity against *E. coli*, *S. aureus*, and *C. albicans*. The release study revealed that almost 99% of Ag was retained on the samples after being in physiological saline solution for 24 h, indicating the good stability of the samples. Samples synthesized in the presence of *P. spina-christi*, *J. regia*, and *S. nigra* leaf extract did not show any cytotoxic effect towards HaCaT and MRC5 cells. Keeping in mind the good antimicrobial activity and low cytotoxicity of the developed textile nanocomposites, this research is expected to move one step further towards the in vivo investigations necessary for application in the wound healing of diabetic patients.

## Figures and Tables

**Figure 1 molecules-29-01447-f001:**
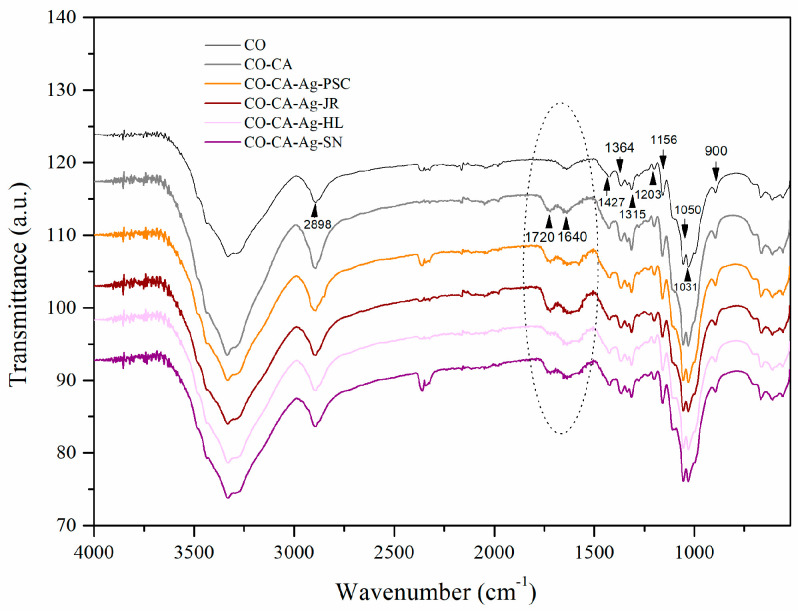
FTIR spectra of CO, CO-CA, CO-CA-Ag-PSC, CO-CA-Ag-JR, CO-CA-Ag-HL, and CO-CA-Ag-SN samples.

**Figure 2 molecules-29-01447-f002:**
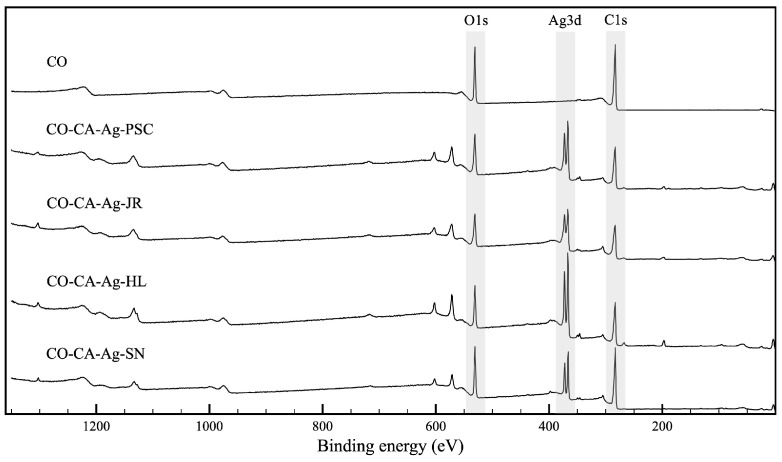
Survey XPS spectra of the samples.

**Figure 3 molecules-29-01447-f003:**
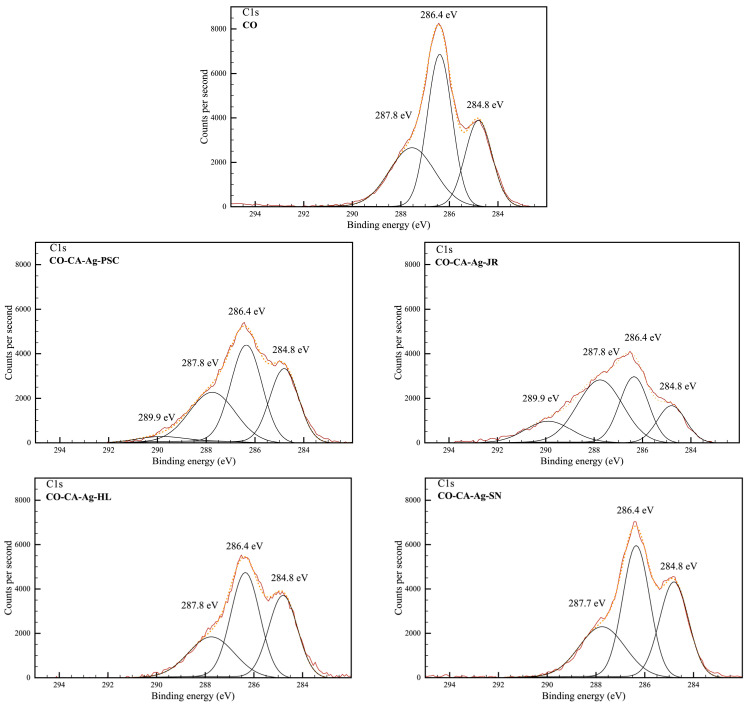
Gaussian deconvolution of high-resolution XPS C1s peaks.

**Figure 4 molecules-29-01447-f004:**
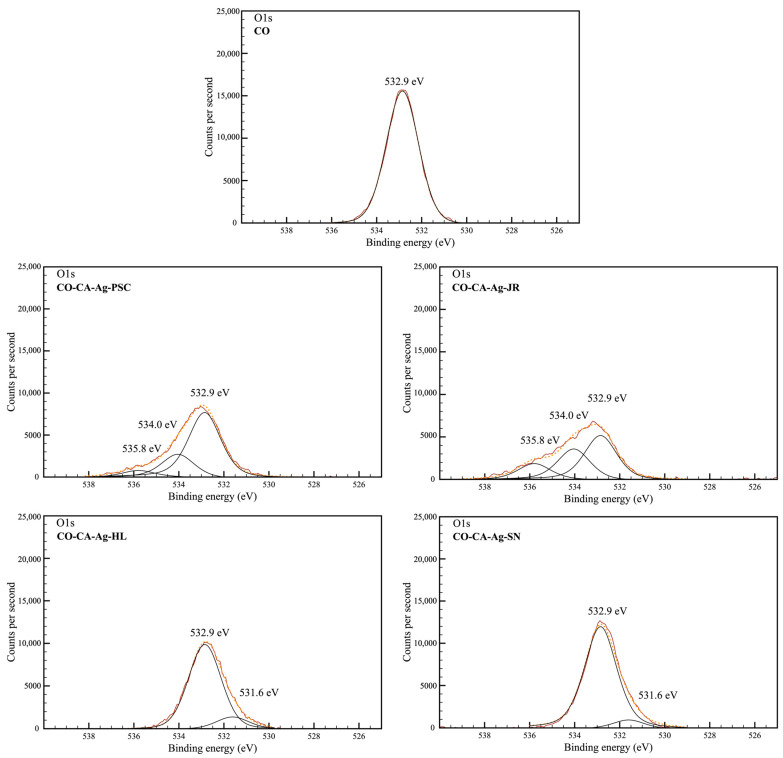
Gaussian deconvolution of high-resolution XPS O1s peaks.

**Figure 5 molecules-29-01447-f005:**
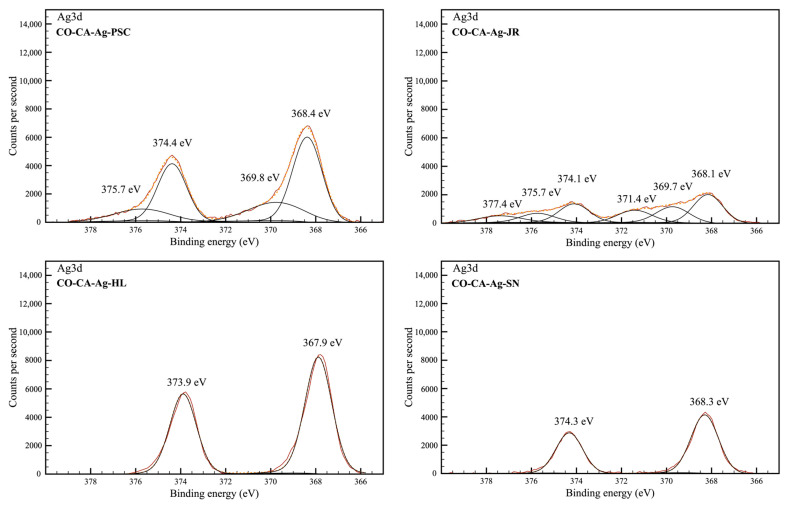
Gaussian deconvolution of high-resolution XPS Ag3d peaks.

**Figure 6 molecules-29-01447-f006:**
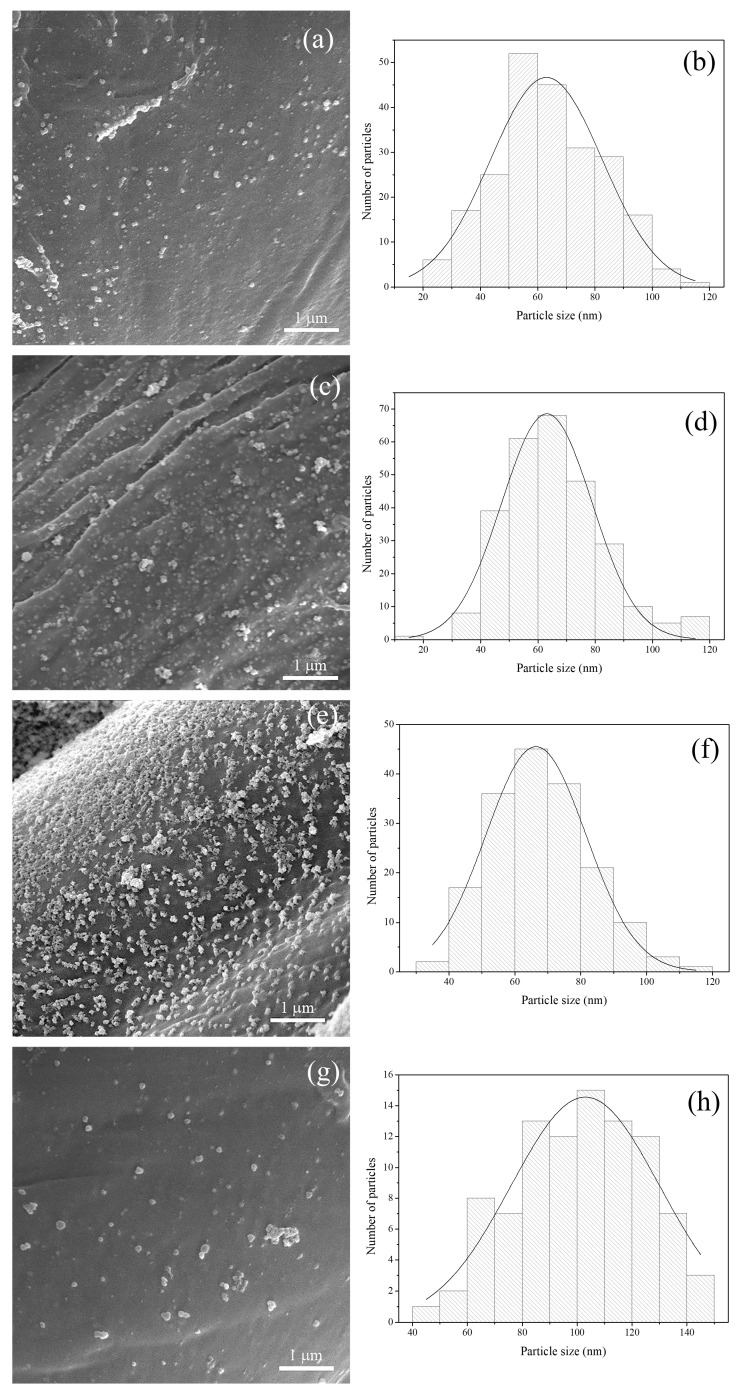
FE-SEM micrographs and size distribution with Gaussian curve of NPs: (**a**,**b**) CO-CA-Ag-PSC, (**c**,**d**) CO-CA-Ag-JR, (**e**,**f**) CO-CA-Ag-HL, and (**g**,**h**) CO-CA-Ag-SN samples.

**Figure 7 molecules-29-01447-f007:**
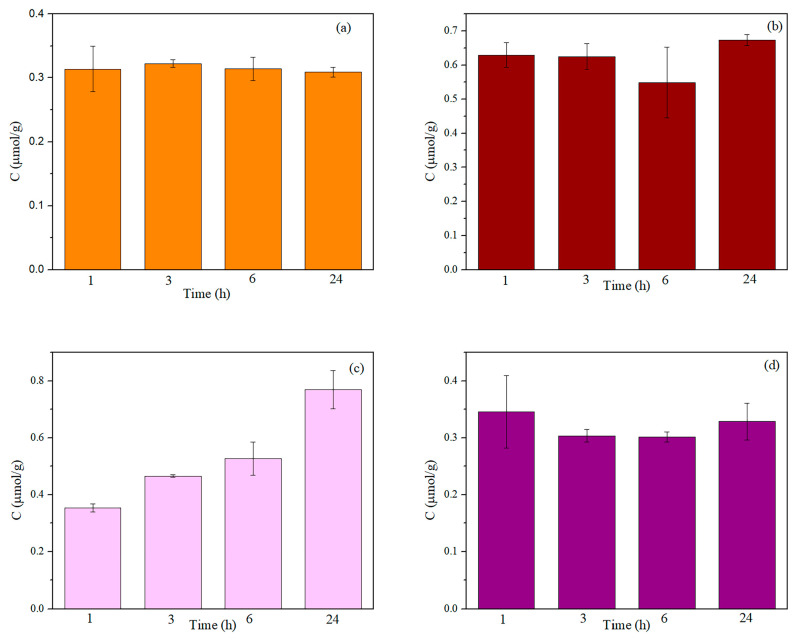
Release of Ag^+^-ions from nanocomposites: (**a**) CO-CA-Ag-PSC, (**b**) CO-CA-Ag-JR, (**c**) CO-CA-Ag-HL, and (**d**) CO-CA-Ag-SN into physiological saline solution.

**Figure 8 molecules-29-01447-f008:**
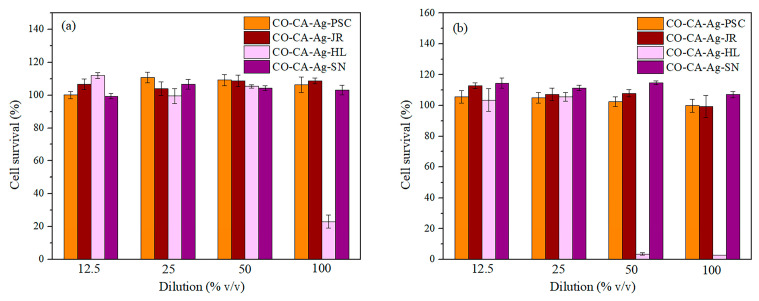
Cytotoxicity of nanocomposites towards (**a**) HaCaT and (**b**) MRC5 cells.

**Figure 9 molecules-29-01447-f009:**
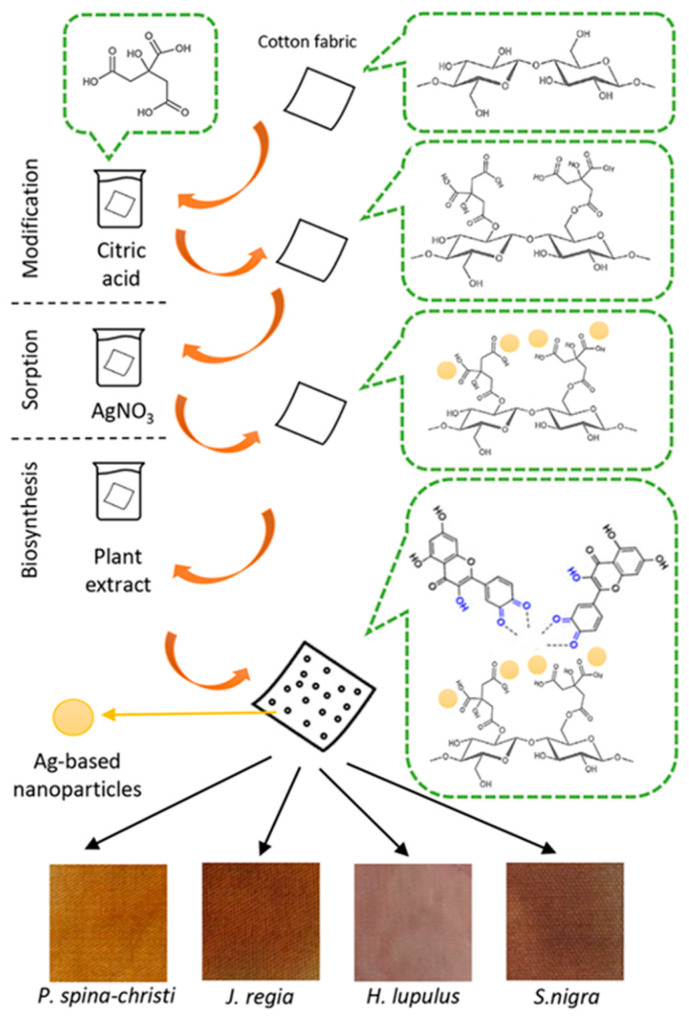
Schematic presentation of synthesis route of nanocomposites.

**Table 1 molecules-29-01447-t001:** Chemical composition of plant extracts.

Compound	Plant (mg/g DW)			
	*P. spina-christi*	*J. regia*	*H. lupulus*	*S. nigra*
Neochlorogenic acid	/	8.27 ± 0.23	/	4.65 ± 0.17
Protocathehuic acid	2.61 ± 0.31	/	1.78 ± 0.03	/
Gallic acid	3.76 ± 0.34	1.20 ± 0.08	0.26 ± 0.01	2.30 ± 0.10
Chlorogenic acid	/	6.72 ± 0.20	/	43.99 ± 0.85
Ellagic acid	/	5.02 ± 0.16	/	/
Vanilic acid	0.80 ± 0.11	/	/	/
Quercetin derivative	6.90 ± 0.21			
Rutin	19.56 ± 0.67	20.63 ± 1.02	2.22 ± 0.10	45.61 ± 0.89
Hyperoside	/	4.06 ± 0.33	0.32 ± 0.02	2.77 ± 010
Isoquercetin	4.12 ± 0.11	3.12 ± 0.22	/	/
Miquelianin	/	8.30 ± 0.31	/	3.22 ± 0.13
Kaempferol-3-*O*-glucuronide	/	6.51 ± 0.19	/	/
Quercitrin	/	5.54 ± 0.13	/	5.61 ± 0.12
3,5-di-*O*-caffeoylquinic acid	/	/	/	1.16 ± 0.08
Kaempferol-3-*O*-glucoside	0.29 ± 0.01	1.21 ± 0.10	0.18 ± 0.01	/
Quercetin *	19.63 ± 0.54	33.22 ± 1.30	1.35 ± 0.08	49.07 ± 1.02
Kaempferol *	0.56 ± 0.07	8.92 ± 0.56	1.22 ± 0.10	0.33 ± 0.02
Isorhamnetin *	/	/	/	2.99 ± 0.09
Total flavonoid content (%)	2.13 ± 0.11	3.95 ± 0.15	0.29 ± 0.03	4.95 ± 0.10
Total polyphenol content (mg GAE/g ext.)	492.93 ± 3.05	220.74 ± 2.11	61.41 ± 0.67	127.40 ± 1.11

* After hydrolysis.

**Table 2 molecules-29-01447-t002:** Element atomic percentages identified on the surface of the samples from the XPS survey spectra.

Sample	C	O	Ag	Cl	Mg	Ca	N	S	P	C/O	C/N
CO	62.72	37.28	-	-	-	-	-	-	-	1.68	-
CO-CA-Ag-PSC	61.13	31.28	4.61	1.84	-	-	-	1.13	-	1.95	-
CO-CA-Ag-JR	61.63	33.57	2.49	2.3	-	-	-	-	-	1.84	-
CO-CA-Ag-HL	57.17	29.83	4.29	3.82	-	-	4.88	-	-	1.92	11.72
CO-CA-Ag-SN	60.83	31.48	1.95	0.56	-	-	4.18	1.0	-	1.93	14.55

**Table 3 molecules-29-01447-t003:** Element atomic percentages identified on the surface of the samples from the XPS survey spectra after sputtering cleaning.

Sample	C	O	Ag	Cl	Mg	Ca	N	S	P	C/O	C/N
CO	78.29	21.41	-	-	-	-	-	-	-	3.66	-
CO-CA-Ag-PSC	63.76	22.77	6.44	1.76	2.1	1.49	-	-	1.67	2.80	-
CO-CA-Ag-JR	66.58	21.23	6.26	2.1	2.61	1.21	-	-	-	3.14	-
CO-CA-Ag-HL	62.89	20.65	8.31	3.92	2.45	1.78	-	-	-	3.05	-
CO-CA-Ag-SN	71.31	21.24	3.96	0.47	1.52	0.93	-	0.57	-	3.36	-

**Table 4 molecules-29-01447-t004:** XPS analysis of C1s and O1s deconvolution spectra in percentages to identify the chemical groups on the surface of the samples.

	C1s				O1s			
	C-C/C-H	C-O	O-C-O	O-C=O	C=O	O-C-O/ C-OH	O-C=O	H_2_O
Samples	284.8 eV	286.4 eV	287.8 eV	289.9 eV	531.6 eV	532.9 eV	534.0 eV	535.8 eV
CO	26.5	43.0	30.5	-	-	100.0	-	-
CO-CA-Ag-PSC	28.4	39.5	29.2	2.9	-	69.2	24.0	6.7
CO-CA-Ag-JR	16.0	28.2	41.8	14.1	-	49.1	33.6	17.3
CO-CA-Ag-HL	33.2	41.9	24.9	-	12.2	87.8	-	-
CO-CA-Ag-SN	32.4	41.2	26.4	-	8.2	91.8	-	-

**Table 5 molecules-29-01447-t005:** XPS analysis of Ag3d deconvolution spectra in percentages.

	Ag^0^/Ag^+^	Ag^2+^	Ag^+^-Ag^2+^	Ag^0^/Ag^+^	Ag^2+^	Ag^+^-Ag^2+^
Samples	367.9–368.4 eV	369.7–369.8 eV	371.4 eV	373.9–374.4 eV	375.7 eV	377.4 eV
CO	-	-	-	-	-	-
CO-CA-Ag-PSC	42.83	17.43	-	28.63	11.10	-
CO-CA-Ag-JR	28.65	17.78	14.27	19.25	10.41	9.64
CO-CA-Ag-HL	59.50	-	-	40.50	-	-
CO-CA-Ag-SN	59.81	-	-	40.19	-	-

**Table 6 molecules-29-01447-t006:** Antimicrobial activity of samples with Ag-based NPs.

Sample	Microorganisms					
	*E. coli*		*S. aureus*		*C. albicans*	
	CFU	R (%)	CFU	R (%)	CFU	R (%)
Inoculum	3.6 × 10^5^		1.5 × 10^5^		1.3 × 10^5^	
Control CO	4 × 10^5^		9.7 × 10^4^		1.1 × 10^5^	
CO-CA-Ag-PSC	850	99.8	3	99.9	705	99.4
CO-CA-Ag-JR	3 × 10^3^	99.2	15	99.9	970	99.1
CO-CA-Ag-HL	580	99.8	20	99.9	750	99.3
CO-CA-Ag-SN	3 × 10^3^	99.2	25	99.9	500	99.5

## Data Availability

The data are contained within the article and Appendix A.

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
