# Peer review of "Plant-Assisted Synthesis of Ag-Based Nanoparticles on Cotton: Antimicrobial and Cytotoxicity Studies"

_molecules, 2024, doi:10.3390/molecules29071447_

Round 1
Reviewer 1 Report
Comments and Suggestions for Authors
Authors should revise the manuscript.

Comments on the Quality of English LanguageMinor issues.
Author Response
Dear Reviewer,
Please see the attachment.
Best regards,
Darka Markovic

Reviewer 2 Report
Comments and Suggestions for Authors
Report on the manuscript molecules-2888873-peer-review-v1 entitled “Plant-assisted synthesis of Ag-based nanoparticles on cotton”. The submitted manuscript should be revised. This work represented the synthesis of Ag-based nanoparticles with the assistance of plant extracts through the application of Paliurus spina-christi, Juglans regia, Humulus lupulus and Sambucus nigra leaves extracts. The presence of nanoparticles with an average size ranging from 57-99 nm on the fiber surface was confirmed. In short, the following points should be addressed:
1. The title should be modified to have the antimicrobial application and cytotoxicity to indicate what could be studied.
2. The language of the manuscript should be checked.
3. In the introduction, a new paragraph about recent materials on cotton should be supported such as the following literatures [Chemical Engineering Journal 476 (2023): 146839 & Molecules, 2021, 26(16), 4731 & Nature Reviews Bioengineering 1, no. 3 (2023): 159-159 & Journal of Colloid and Interface Science, 2020, 580, pp. 822–833].
4. XPS fitting or figure 3 should be improved.
5. What about TGA and BET analysis?, they could support the characterization.
6. What about Repeatability and reproducibility as key indicators of the quality of experiments and analyses? Do the authors repeat their data 3 times or more to confirm reproducibility especially for antimicrobial activities?
Comments on the Quality of English LanguageIt should be revised.
Author Response

(The authors gave the same response as above.)

Reviewer 3 Report
Comments and Suggestions for Authors
see attachment

Comments on the Quality of English Languageminor English editing.
Author Response

(The authors gave the same response as above.)

Round 2
Reviewer 2 Report
Comments and Suggestions for Authors
Accepted
Comments on the Quality of English LanguageAccepted
Reviewer 3 Report
Comments and Suggestions for Authors
Authors have addressed most of the comments. However, a few comments needs to be considered.
- although comment #1 of using different colors for the fitted XPS spectra was not answered and it can be fine, other comments regarding the update of literature on plant-extract based Ag NPs by considering recent articles like the suggested other two 10.1039/D3RA05070J and 10.1016/j.ijbiomac.2023.128009 should be included as they relevant to the topic.